# Associations of Social and Psychological Resources with Different Facets of Chronic Stress: A Study with Employed and Unemployed Adolescents

**DOI:** 10.3390/ijerph17145032

**Published:** 2020-07-13

**Authors:** Laura M. Wade-Bohleber, Carmen Duss, Aureliano Crameri, Agnes von Wyl

**Affiliations:** 1Psychological Institute, Zurich University of Applied Sciences, 8037 Zurich, Switzerland; carmenduss@hotmail.com (C.D.); cman@zhaw.ch (A.C.); vonw@zhaw.ch (A.v.W.); 2Department of Psychiatry, Psychotherapy, and Psychosomatics, University Hospital of Psychiatry, 8032 Zurich, Switzerland

**Keywords:** adolescence, unemployment, chronic stress, social resources, psychological resources

## Abstract

Adolescents navigate many psychosocial changes. A critical transition in adolescence is the one from school to work life. Both taking the first steps in work life and the failure to achieve this transition and being unemployed can engender elevated levels of stress during adolescence. Stress, especially when experienced chronically, is an important risk factor for mental health problems. Social and psychological resources may mitigate the experience of chronic stress. This study explored associations of social and family support, self-esteem, and self-efficacy with different dimensions of chronic stress in a sample of 1405 employed and unemployed adolescents (*M*(age) = 17.84, *SD* = 1.63, range: 14.05–26.12) in Switzerland. Unemployed adolescents showed higher stress levels overall. Higher levels of social and psychological resources were generally linked to lower stress levels. Social support and self-esteem predicted stress levels most consistently and strongly. On several stress dimensions, the association between higher self-esteem and lower stress levels was more pronounced in employed youth whereas the association between higher social support and lower stress levels was stronger in unemployed youth. Our findings provide insights on the differential associations of social and psychological resources with various facets of chronic stress in the context of employment and unemployment during adolescence.

## 1. Introduction

### 1.1. The Transition of School to Work Life and the Experience of Stress during Adolescence

Adolescence, with its many biological and psychosocial transitions, is a vulnerable period for the development of mental disorders [1]. From a developmental perspective, it is a time of high exposure to risk factors with a limited availability of psychological and social resources [2]. It has been suggested that exposure to stress is an influential factor in determining an individual’s vulnerability to developing a mental disorder during adolescence [3]. Overall, the detrimental effects of chronic stress on mental health via neuroendocrinological pathways are well documented [4,5] and there is evidence that the changing adolescent brain may be particularly sensitive to stress [6].

A critical transition in adolescence is the one from school to work life. In Switzerland, the majority of adolescents (62% in 2019 [7]) engage in an two- to four-year apprenticeship after lower secondary education, which most of them finish at the age of 15 years. This implies that these adolescents have to choose a profession and adapt to a new role and an adult work rhythm at a young age—factors that may contribute to experiences of increased stress. In fact, evidence suggests that adolescent employees in Switzerland exhibit elevated levels of chronic stress [8].

Some adolescents fail to accomplish the transition from school to work life and, in consequence, face unemployment. In Switzerland, the unemployment rate of 15–24-year-olds was 8% in 2019 [9]. In comparison with other European countries, this rate is low, yet unemployment may be experienced as particularly stigmatizing [10,11] as Switzerland is a country with a strong work ethic and a notoriously low unemployment rate (overall unemployment rate of 4.2% in the first quarter of 2020 [12]). Unemployed youth are eligible to receive financial support by the Swiss unemployment insurance after a certain waiting period and for a limited amount of time [13]. This unemployment insurance also provides different support offers for adolescents seeking employment. For instance, many adolescents who leave school without the prospect of a job or an apprenticeship participate in a so-called transitional program, which provides an occupational curriculum with educational, professional and coaching elements.

In all age groups, unemployment can engender chronic stress as it continuously affects an individual’s daily life routines [14] and is associated with poor mental health outcomes [15,16]. Meta-analytical evidence suggests that unemployment is not only correlated with mental distress but actually causes it [17]. During adolescence, experiencing unemployment has particularly detrimental effects, impacting mental health well into adulthood [18,19,20,21].

### 1.2. Stress as an Imbalance of Demands and Resources

Stress has been conceptualized in different ways, e.g., as a physiological response pattern of the organism as part of an adaptation effort to an environmental challenge [22], as a stimulus to which an individual needs to adapt and which can be characterized by different qualities and typical reactions [23], or as a dynamic process resulting from a transaction of an individual and his complex environment [24]. According to this latter viewpoint, stress results from factors in the environment perceived as stressful by an individual, depending on their coping mechanisms. Expanding this transactional conceptualization of stress, the systemic demands–resources (SDR) model of health [25] understands health as a result of complex and dynamic adaptation and regulation processes between an individual and their environment. The specific merit of the SDR model is the conceptualization of both internal and external demands and resources that can also contribute to a differentiated understanding of the experience of stress. External demands have their origin in the individuals’ environment, such as expectations or claims that are made by significant others or constraints caused by the work situation of the individual. Internal demands result from expectations, beliefs, and moral values of an individual that concern themselves and their environment. External resources are provided by the environment such as social support, education, and financial capital. Internal resources refer to intrapsychic characteristics such as self-esteem and the sense of self-efficacy. An individual experiences stress when external or internal demands exceed his or her external and internal resources [24,26]. This experience can become chronic if frequent or continuous [27]. Chronic stress is a substantial risk factor for mental health problems [28] with more pronounced negative effects on health than the experience of single stressful events [28]. Consistent with the SDR model, the Trier Inventory of Chronic Stress (TICS) [29,30] allows measurement of different facets of chronic stress that result from internal and external demands (cf. description of the instrument in Section 2).

Unemployment tends to be accompanied by a deterioration in external and internal resources. Jahoda [31] argued that unemployed individuals lack both manifest, i.e., intended like those linked to financial income, and latent, i.e., unintended, benefits linked to psychological needs such as time structure, social contact, common goals, status, and activity provided by employment. At the same time, external (e.g., the expectation by the entourage to be a working member of society) and internal (e.g., expectations of oneself to be successful in one’s career) demands likely increase, contributing to an imbalance of resources and demands. In line with the SDR model, adaptation to unemployment and individual stress levels may particularly depend on how much internal and external resources are affected by change in employment status. This line of conceptualization has evident parallels with Hobfoll’s Conservation of Resources Theory [32], which has also previously been used to conceive adaptation to unemployment in young adults [33].

### 1.3. External and Internal Resources Mitigating the Effects of Stress

In stress research, three external and internal resources have emerged as particularly beneficial to an individual when experiencing stress: social support, self-esteem, and a sense of control over life or self-efficacy [28].

An individual’s social support is the availability of emotional, instrumental, and informational support and social companionship provided by significant others [34]. The stress-mediating effects of social support are well documented for adults [35], yet findings referring to adolescence are somewhat more mixed [36,37]. In this developmental period, the effectiveness of stress buffering by social support may depend on their source, e.g., if support is provided by parents or peers [38].

There is a broad literature on the stress-buffering effects of social support during unemployment [39]. More specifically, several studies have demonstrated the association of social support with positive mental health outcomes in unemployment during adolescence. For instance, in a study with more than 8000 unemployed youth in Northern Europe, Hammer [40] showed that parental and peer support were important predictors of mental health. Similarly, Bjarnason and Sigurdardottir [41] found that parental emotional support impacts psychological distress in unemployed youth. It has also been suggested that family support may compensate for the absence of some of the latent benefits of employment [42].

Self-esteem and self-efficacy can be conceived as components of so-called core self-evaluations. These are fundamental assumptions an individual makes about her- or himself and her or his agency within her or his environment [43]. More specifically, global self-esteem consists of the individual’s positive or negative attitudes toward the self and can be distinguished from specific self-esteem relating to attitudes towards the self regarding distinct contexts such as academia [44]. General self-efficacy is the belief in one’s competence to effectively deal with stressors and produce certain outcomes and can, also, be distinguished from specific forms of self-efficacy referring to one’s performance at specific tasks [45,46]. Self-esteem and self-efficacy have been identified as important internal resources affecting resilience, i.e., the capacity to successfully adapt to a changing environment, during adolescence [47,48,49,50,51]. Global self-esteem and general self-efficacy buffer the effects of stress in adolescence [52,53], however some studies have also reported contradictory findings [54].

Meta-analytic evidence also suggests that core self-evaluations such as self-esteem and self-efficacy are associated with mental health outcomes and life satisfaction during unemployment [55]. For instance, young adults who can conserve a sense of environmental mastery, i.e., a sense of being able to influence or impact their environment, show lower levels of psychological distress during unemployment [33], and this is consistent with findings in unemployed adults [56].

### 1.4. Differential Effects of Internal and External Resources in Employment and Unemployment

The beneficial effects of internal and external resources on mental health may act through direct or indirect pathways. Most prominently in stress research, two models describe different mechanisms of action of social support [34,57]: the main effect model proposes that social relationships are beneficial to an individual’s mental health whether or not they experience stress; the stress-buffering model posits that social relationships are mainly beneficial for individuals in the context of experiencing stress. These models can also be applied to conceive the pathways by which internal resources affect mental health: self-esteem and self-efficacy may also be beneficial to an individual’s mental health regardless of their stress levels (main effect model) or they may take a specific buffering effect when an individual experiences stress (stress buffering model). These two models are not mutually exclusive [34,57]. For instance, it has been demonstrated that adults benefit from elevated levels of social support both when being employed and unemployed [39], the latter being a source of increased stress [17].

In the context of our study, the transition of school to work life may engender increased levels of chronic stress in adolescence [8], yet facing unemployment at this period of life is likely associated with particularly high levels of chronic stress [17,41]. Comparing both groups, employed and unemployed adolescents, we may observe both main effects and stress buffering effects of internal and external resources. Hereby, the subjective experience of different facets of chronic stress can serve as an indicator of individual adaptation to this transitional period.

### 1.5. Aims and Hypotheses of the Current Study

To our best knowledge, no study has examined the specific associations of external and internal resources with the experience of chronic stress in adolescence in the context of the transition from school to work and the respective failure to accomplish this step. More detailed information is needed on what factors likely contribute to a successful adaptation to this critical transition in adolescence, especially in the Swiss context as the insertion into the labor market happens early in age compared to other high-income countries.

The aim of this study was thus to explore the interplay of social and family support, self-esteem and self-efficacy with different dimensions of chronic stress in a sample of adolescents with and without employment in Switzerland. We hypothesized that higher levels of perceived social and family support, and higher self-esteem and self-efficacy, would be associated with lower stress levels on all stress dimensions (main effect model). We also expected to find higher stress levels in unemployed youth compared to employed youth. We were interested in whether external and internal resources moderate the association of employment status and the different facets of stress. We expected to see more pronounced negative associations of external and internal resources with stress levels in unemployed youth (stress buffering model).

## 2. Materials and Methods

### 2.1. Participants

Participants of this study were part of a larger stress prevention study in adolescence [58] and constitute a convenience sample. As the original study comprised more employed than unemployed adolescents, we recruited an additional 308 unemployed adolescents for the study reported here. This resulted in a sample size of 1405 participants comprising 906 employed and 499 unemployed adolescents. Inclusion criteria were (a) for employed adolescents to be in their first year of apprenticeship; (b) for unemployed adolescents to participate in a transitional program—a specific state- or NGO-run offer for youth seeking an apprenticeship or employment; (c) sufficient language skills to fill in questionnaires. Demographic characteristics of the sample are described in Table 1.

### 2.2. Procedure

Employed adolescents were recruited in a large Swiss company in different regions of the German part of Switzerland. This Swiss company participated in the original stress prevention study [58]. Data were collected at the end of the first year of their apprenticeship. Apprenticeships covered a wide range of jobs in different fields such as business and administration, retail, mechanics, logistics, and IT. Unemployed adolescents were recruited from different transitional programs for unemployed youth across the German part of Switzerland. These transitional programs offer an intermediate occupational solution for youth without employment and are mostly run by the state or non-profit organizations. Adolescents can only access these transitional programs if they are unemployed. Transitional programs were contacted by the study team to evaluate interest in a study participation on a voluntary basis. Data were collected when unemployed adolescents joined the transitional program. All participants, employed and unemployed, were informed about the study by members of the study team in form of a group presentation or by social workers working in the transitional programs. Participation was voluntary. Participants filled in paper and pencil questionnaires. This study collected anonymized data only and, according to Swiss legislation (Swiss Human Research Act), did thus not require approval from the local ethics committee.

### 2.3. Instruments

#### 2.3.1. Stress Dimensions

The Trier Inventory of Chronic Stress (TICS [29,30]) measures different dimensions of stress. The scales work overload (OVRL) and social overload (SOVRL) capture stress that results from high demands from the individual’s environment (external demands). The scales occupational discontent (DIS), lack of social recognition (RECO), and social isolation (ISO) relate to stress caused by a lack of satisfaction of the individual’s needs (internal demands). The TICS scales can be used to assess the individual’s situation in a work environment but can also refer to tasks of daily life. An additional scale measures chronic worrying (WRY). The TICS also comprises a screening scale of chronic stress (SCR) using the 12 of the most salient items of all scales. Table 2 provides an outline and example items of the different TICS scales. Items are rated on a 5-point Likert scale (0 = never to 4 = very often) with reference to the experience of stress during the previous three months. In our sample, all scales of the TICS showed acceptable to excellent internal consistency (Cronbach’s alpha range was 79–91).

#### 2.3.2. External (Social) Resources

Satisfaction with social support (SocS) was measured using a four-item scale of the social support questionnaire (*Fragebogen zur Sozialen Unterstützung*, F-Soz-U [59]). This is a questionnaire widely used in German speaking countries. Items such as “I don’t know enough people who I can ask for help when I encounter a problem” (recoded) are rated on a 5-point Likert scale (0 = strongly disagree to 4 = strongly agree). This scale showed good internal consistency (Cronbach’s α = 84).

Family support (FamS) was captured by four items of the quality of life in children and adolescents inventory (*Inventar zur Erfassung der Lebensqualität bei Kindern und Jugendlichen*, ILK [60]). These items refer to how the adolescents feel in regards to their relationship with their parents (e.g., “I’ve been getting along well with my parents”) and are rated on a 5-point Likert scale (0 = never to 4 = always). Cronbach’s alpha for FamS was 77 (acceptable).

#### 2.3.3. Internal (Psychological) Resources

Self-esteem (SelfE) was measured by four items of the Rosenberg self-esteem scale [61], one of the most frequently used self-esteem scales. The items evaluate statements about the individual such as “On the whole, I am satisfied with myself” and are answered on a 5-point Likert scale (0 = strongly disagree to 4 = strongly agree). Internal consistency for SelfE was acceptable (Cronbach’s α = 77).

A general sense of perceived self-efficacy (GSE) was captured by The General Self-Efficacy Scale (GSE, *Skala zur Allgemeinen Selbstwirksamkeitserwartung* [62]), which comprises 10 items. An example item is “I can solve most problems if I invest the necessary effort”. Responses are given on a 4-point Likert scale (0 = not at all true to 4 = exactly true). Cronbach’s alpha for GSE was 86 (good).

### 2.4. Statistical Analyses

All statistical analyses were conducted using SPSS 24 (IBM Corp. Released 2016. IBM SPSS Statistics for Macintosh, Version 24.0. Armonk, NY, USA: IBM Corp.).

We tested if data were missing at random using Little’s MCAR test. Little’s MCAR test performed on all scales (SCR, OVRL, SOVRL, DIS, RECO, ISO, WRY, SocS, FamS, SelfE, GSE), age, and sex revealed was not significant, with *X^2^* (646, *N* = 1405) = 644.34, *p* = 0.51. Consequently, we did not reject the null hypothesis assuming that the data were missing completely at random. We replaced missing values in metric variables using linear interpolation in SPSS. Fifteen participants did not report their sex. We therefore performed analyses with complete datasets of 1390 participants.

We used descriptive statistics to describe the sample, and Pearson product-moment correlation of all metric variables to get a first impression of the data (cf. Appendix A). Multiple linear regression then served to explore associations of different stress dimensions with employment status and external and internal resources. The different TICS stress scales served as dependent variables. We used nested linear models to test how much variance was explained by different predictors. Age and sex were potential confounding variables and entered into the model first. Then, we consecutively entered employment status, external resources (social support, family support), and internal resources (self-esteem, self-efficacy). In line with Gelman [63], numeric predictors were divided by two times their standard deviations in order to be able to compare them with the binary predictors (employment status, sex). In order to determine moderation effects, we tested the interactions of employment status with the following variables: social support, family support, self-esteem, and self-efficacy. We first mean-centered these variables, built the interaction terms, and then entered them into nested models.

We visually inspected residual plots of all regression models which revealed no obvious deviations from homoscedasticity or normality. Multicollinearity of predictors was not a concern, with the variance inflation factor (VIF) mostly <2. For a few exceptions, mostly the interaction terms, VIF was higher but never >4 in any case.

## 3. Results

Correlation analysis revealed that most scales measuring external (social and family support) and internal (self-esteem and self-efficacy) resources were negatively associated with stress dimensions (cf. Appendix A). Age weakly correlated with some of the variables of interest (mostly positively with stress dimensions and negatively with external and internal resources, yet effect sizes were very small).

Nested regression models are described in Table 3, Table 4, Table 5 and Table 6. Employment status, social and family support, self-esteem, and self-efficacy consistently predicted chronic stress as measured by the screening scale (SCR) and specific facets of chronic stress such as chronic worrying (WRY), work overload (OVRL), occupational discontent (DIS), and social isolation (ISO). Employment, higher social and family support, higher self-esteem and higher self-efficacy were associated with lower stress levels, with one exception: employment was associated with higher levels of stress due to work overload (OVRL).

Employment status, social and family support, and self-esteem, but not self-efficacy, predicted stress due to a lack of social recognition (RECO). Employment was associated with lower stress levels on these scales. Higher social and family support, and higher self-esteem, were associated with lower stress levels on these scales.

Employment status was not associated with stress due to social overload (SOVRL), but social and family support, self-esteem, and self-efficacy were. Higher social and family support and higher self-esteem and self-efficacy predicted lower stress due to social overload.

Concerning the distinction of stress resulting from external (OVRL, SOVRL) or internal (DIS, RECO, ISO) demands, we did not observe any patterns of differences in associations with external and internal resources.

The standardized coefficients in the different regression models indicated that, overall, social support and self-esteem were the strongest predictors of stress levels. Further, age and sex as confounding variables explained little variance in the regression models, while the variance explained by external and internal resources together was between 19.4% (SOVRL) and 44% (ISO).

We observed that external and internal resources moderated the association of employment status and several stress dimensions. These moderation effects are illustrated in Figure 1. The negative association of self-esteem and different facets of stress were more pronounced in employed youth: we observed this for chronic stress (SCR), stress due to work overload (OVRL), stress due to occupational discontent (DIS), and stress due to a lack of social recognition (RECO). In contrast, the negative association of social support and two facets of stress were more pronounced in unemployed youth: we observed such effects for stress due to a lack of social recognition (RECO), and social isolation (ISO). However, it is important to note that adding these moderation terms to our regression models only led to a small increase in explained variance (between 0.3% (SCR) and 2.9% (SOVRL)).

## 4. Discussion

The aim of this study was to explore associations of employment status and external (social) and internal (psychological) resources with different facets of chronic stress in adolescents with and without employment. Moreover, we were interested in examining if the association of employment status and different facets of chronic stress was moderated by external and internal resources.

We observed that employment was generally associated with lower levels of chronic stress. There was one exception to this: employment was associated with higher levels of stress due to work overload (OVRL). Higher external (social and family support) and internal (self-esteem and self-efficacy) resources mostly predicted lower levels of chronic stress on all dimensions.

We found different moderation effects of external and internal resources in the association of employment status and the different facets of chronic stress. The association of higher self-esteem and lower stress levels was more pronounced in employed youth than in unemployed youth across several dimensions of stress. On the other hand, the negative association of social support and two stress dimensions (lack of social recognition (RECO) and social isolation (ISO)) were more pronounced in unemployed youth compared to employed youth.

### 4.1. Overall Increased Levels of Chronic Stress in Unemployment and Increased Chronic Stress Due to Work Overload in Employment

Our findings of increased levels of chronic stress in the unemployed group fit well into the existing literature demonstrating deteriorated mental health in unemployed youth [19,20,64]. The link between chronic stress and negative mental health outcomes has been subject to intense research efforts and a large body of evidence now demonstrates the neuroendocrinological, immunological, and metabolic pathways involved [65,66]. Effects of chronic stress may be particularly detrimental in adolescence [6], given the major changes in physiological systems and the brain during this developmental period [67,68]. From a psychosocial perspective, adolescence is a sensitive period of reorientation, in which young individuals have to negotiate who they are and who they want to be [69,70]. Experiencing unemployment interferes with aspects of identity negotiation such as forming an occupational identity, which has been related to more negative mental health outcomes in youth [71]. This aspect resonates with Jahoda’s [31] conception of a lack of latent benefits such as social status provided by employment and which is thought to have negative effects on mental health.

Interestingly, employed youth showed higher levels of stress due to work overload (OVRL). We presume that higher levels of stress on this dimension are associated with the circumstances of entering the labor market. The employed youth participated in our study at the end of their first year of apprenticeship in a big Swiss company. Leaving secondary school and starting an apprenticeship is associated with many changes in daily life such as working a 42- to 45-h week, integrating into new structures of a company, and assuming new responsibilities linked to their professional position. Many Swiss apprentices report significant levels of stress and to a higher extent than their older colleagues [8,72], which may also mirror a difficulty of adjustment to these changes.

### 4.2. Associations of External and Internal Resources and Different Facets of Chronic Stress

Overall, we found that both higher external and higher internal resources showed negative associations with different facets of chronic stress. This aligns with the large body of research demonstrating the protective effects of social and psychological resources during adolescent development [2,73,74]. For instance, in a study with 2860 adolescents, Wille et al. [2] found that self-worth, family and social support served as protective factors when assessing mental health problems. Self-worth is conceptually close to self-esteem [75]. Our findings thus align with those of Wille et al. [2] concerning the role of self-worth/self-esteem, family, and social support. However, there was also one contradicting finding: while Wille et al. [2] observed that self-efficacy was not a protective factor, our study associated higher self-efficacy with lower stress levels on most dimensions. This may be linked to the different outcome measures used: Wille et al. [2] used the Strengths and Difficulties Questionnaire [76] capturing behavioral, emotional, and peer problems of adolescents. In our study, we focused on chronic stress. Self-efficacy likely supports adaptive strategies of appraisal in the experience of stress [77] and mitigates the negative effects of stress during adolescence [53]. However, recent evidence suggests that the relationship between self-efficacy, the experience of stress, and mental health outcomes may be more complex than originally thought [78,79,80].

Interestingly, overall, the perception of social support and self-esteem were more consistent and stronger predictors of chronic stress dimensions than employment status, family support, and self-efficacy. This aligns with the fact that these two factors have been explored most consistently across diverse studies in their function as protective agents for mental health during adolescence but also during the life span [44,81,82] and for the effects of unemployment [55]. In a study of nearly 8000 youth across different countries in Scandinavia, Hammer [40] also found that internal resources such as individual coping strategies predicted mental health stronger than the employment status per se. Another study demonstrated that neuroticism was a stronger predictor than length of unemployment for psychological distress [83]. A comparison with additional studies is difficult as employment status is rarely used as a predictor of mental health outcomes but, most often, the association of certain variables with mental health outcomes is tested in unemployed populations [55]. However, our specific observation on the strength of the different predictors seems contrary to the large evidence on the major impact of unemployment on psychological distress in particular [41] and mental health in general [20]. One possible explanation for this is that adolescents in our sample may not have experienced extensive periods of unemployment. They may still experience their current unemployment situation as transitional. It is possible that chronic stress in the unemployed group of our study is not a consequence of unemployment but a pre-existing condition. Our cross-sectional study design does not allow us to shed light on the causal influence of unemployment on stress levels. In this context, it is interesting to note that employment status was a strong predictor of two specific dimensions of chronic stress: occupational discontent (DIS) and chronic worrying (WRY). That unemployment is associated with increased stress due to discontent with the occupational situation seems likely. The chronic worrying (WRY) scale captures aspects that are close to depressive and anxious symptoms. Increased chronic worrying of the unemployed group may be indicative of more frequent mental health issues in unemployed compared to employed adolescents.

Conceptionally, our study used the SDR model [25] to conceive the experience of stress and, in consequence, implemented the TICS to measure stress. The TICS differentiates between chronic stress resulting from external (i.e., excessive work load) and internal (i.e., lacking social recognition) demands. It is interesting to note that we did not observe any consistent pattern of associations between external and internal demands and resources (e.g., that higher social support would show stronger associations with lower stress levels that result from external demands). Further research is needed to determine the usefulness of the proposed distinction of chronic stress resulting from external or internal demands.

### 4.3. Differential Associations of External and Internal Resources with Chronic Stress in Employment and Unemployment

One interesting finding of this study is the different moderation effects for self-esteem and social support in unemployed and employed youth. First, in employed youth, the association of higher self-esteem and lower stress levels was more pronounced than in unemployed youth. We observed such moderation effects for chronic stress (SCR) and three other stress dimensions (stress due to work overload (OVRL), occupational discontent (DIS), and lack of social recognition (RECO)). Second, in unemployed youth, the negative association of social support and two stress dimensions (lack of social recognition (RECO) and social isolation (ISO)) was more pronounced.

In light of Cohen and Willis’ [34] conceptions of a main effect and a stress-buffering effect of social support, one may suppose that these moderation effects indicate that self-esteem is beneficial for adolescents regardless of the context of experiencing a critical life event such as unemployment. In this line of thought, social support may be particularly important as a stress buffer when adolescents face unemployment and thus experience increased levels of chronic stress compared to employed adolescents. However, we must address that our original assumption was that not only unemployment, but also a successful transition of school to work life, leads to increased levels of stress in youth.

These moderation effects also indicate that it may be relevant to conceive stress prevention strategies differently for employed and unemployed adolescents: employed adolescents may be able to mobilize internal resources such as their self-esteem, and unemployed youth their external resources such as social support, to cope with the effects of stress. Nonetheless, considering the observed moderation effects, it is important to note that adding these only led to a small increase of explained variance in our regression models (between 0.3% and 2.9%). Thus, the meaning of these findings is limited and the specific role of such moderation effects of external and internal resources on the experience of chronic stress in employment and unemployment in adolescents needs to be further clarified.

### 4.4. Limitations

Our findings are subject to several limitations. First of all, we explored associations of external and internal resources with different facets of chronic stress in a cross-sectional study. Therefore, we cannot draw any causal inferences. A longitudinal study exploring the here-revealed effects could contribute to a better understanding of the causal relationships between external and internal resources and adolescent adjustment in employment and unemployment. Second, we did not test for moderation effects of external and internal resources, although there is evidence that such effects exist (e.g., higher levels of social support may be associated with increased self-esteem [84]). Third, we did no test for interaction effects of sex and unemployment even though males were overrepresented in the unemployed group. Furthermore, there is substantial evidence that mental health is differently affected by unemployment in men and women [15,85], a fact that we did not account for in our analyses. Further, we did not pay sufficient attention to gender identity when designing our questionnaire. Participants had solely the option to answer if they were “female” or “male”. We did not include an additional option for participants who may have felt uncomfortable or incorrectly addressed answering this question. This may have been a reason why we had 15 none-answers to this question. Fourth, and most importantly, we did not take into account other important factors that may moderate the effects of unemployment on an individual and their experience of chronic stress. For instance, McKee-Ryan et al. [55] identified other factors such as work-role centrality, cognitive appraisal, coping strategies, and length of unemployment as contributing to individual well-being after job loss. Length of unemployment may have been a particularly important factor to control for in our study: longer periods of unemployment have more pronounced effects on mental health [86], tend to erode social support [41] and likely also affect the financial resources of adolescents [87], which may additionally contribute to increased stress levels [40,88]. Additionally, we did not take into account the socioeconomic status including financial issues of the adolescents nor of their parents in our sample, even though financial resources seem to play an important role for youth stress levels during unemployment and their psychosocial adjustment to the situation [40,88,89]. Further, previous evidence suggests that the stress-buffering effects of social support also depend on the socioeconomic resources of the individual [57,83]. In Switzerland, unemployment may affect the financial resources of adolescents less than in some other countries as adolescents can apply for financial support through the Swiss unemployment insurance [13]. Furthermore, most of the adolescents live with their parents [90] and families in Switzerland have a high income in comparison to other European countries [91]. However, these general observations do not imply that financial resources do not impact stress levels in adolescents in Switzerland and the interplay of these factors remains to be specifically explored and further studied. Last, we did not consider additional risk factors in either of the groups, such as adolescent or parental physical or mental illness or family status (e.g., growing up with a single parent) even though there is evidence that these are risk factors for adolescent development and may be accompanied by increased levels of chronic stress [2].

In sum, the interplay of external and internal resources and chronic stress in unemployed and employed adolescents merits further study in a longitudinal design, taking into account additional contributing factors such as length of unemployment, socioeconomic status, gender, and parental and adolescent mental impairments.

## 5. Conclusions

To our knowledge, our study is the first to explore different facets of chronic stress and their associations with different individual resources in employed and unemployed adolescents. Our findings shed light on the differential associations of employment status, social and family support, self-esteem and self-efficacy with different facets of chronic stress in adolescence. They also provide insights on divergent strengths of some of these associations in employed and unemployed youth.

## Figures and Tables

**Figure 1 ijerph-17-05032-f001:**
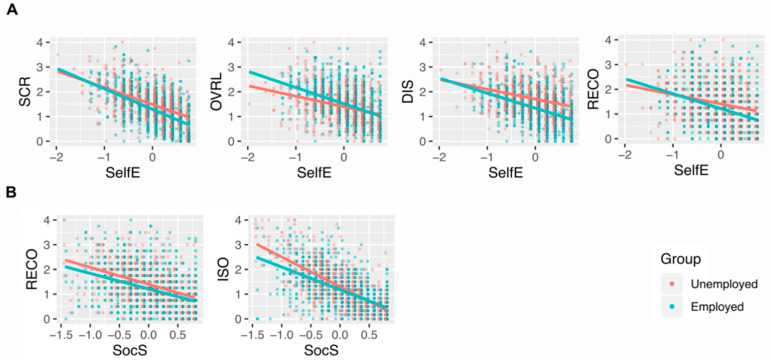
Differential associations of external (social) and internal (psychological) resources with facets of chronic stress in employed and unemployed youth. (**A**) The association of self-esteem (Self E) (FamS) with different facets of chronic stress is more pronounced in employed youth. (**B**) The association of social support (SocS) with different facets of chronic stress is more pronounced in unemployed youth. SCR = chronic stress screening, OVRL = stress due to work overload, DIS = stress due to occupational discontent, RECO = stress due to lack of social recognition, ISO = stress due to social isolation. Data of unemployed adolescents are illustrated in red, data of employed adolescents in green.

**Table 1 ijerph-17-05032-t001:** Demographics.

	Employed	Unemployed
*n*	906	499
	*n* = 894	*n* = 496
Female (%)	491 (44.5)	197 (39.5)
Male (%)	403 (54.2)	299 (59.9)
	*n* = 828	*n* = 462
Age (*M* (*SD*) range)	17.69 (1.50) 14.85–25.70	17.99 (1.74) 14.05–26.12
Stress (TICS): screening scale	*n* = 906	*n* = 499
SCR (*M* (*SD*))	1.26 (0.71)	1.53 (0.72)
Stress (TICS): external demands		
OVRL (*M* (*SD*))	1.49 (0.72)	1.42 (0.7)
SOVRL (*M* (*SD*))	1.33 (0.70)	1.47 (0.76)
Stress (TICS): internal demands		
DIS (*M* (*SD*))	1.31 (0.68)	1.73 (0.69)
RECO (*M* (*SD*))	1.19 (0.83)	1.43 (0.78)
ISO (*M* (*SD*))	1.16(0.78)	1.37(0.90)
Stress (TICS): chronic worrying		
WRY (*M* (*SD*))	1.37 (0.86)	1.78 (0.52)
External resources		
Satisfaction with social support (SocS) (*M* (*SD*))	2.62 (0.89)	2.48 (0.89)
Family Support (FamS) (*M* (*SD*))	3.18 (0.81)	3.04 (0.85)
Internal resources		
Self-esteem (SelfE) (*M* (*SD*))	2.96 (0.68)	2.81 (0.80)
Self-efficacy (GSE) (*M* (*SD*))	1.92 (0.45)	1.79 (0.52)

TICS = Trier Inventory of Chronic Stress, SCR = Chronic stress screening scale, OVRL = stress due to work overload, SOVRL = stress due to social overload, DIS = stress due to occupational discontent, RECO = stress due to a lack of social recognition, ISO = stress due to social isolation, WRY = stress due to chronic worrying.

**Table 2 ijerph-17-05032-t002:** Different scales of the Trier Inventory of Chronic Stress (TICS).

Stress Dimension and Associated TICS Scales	Scale Abbrevation	Number of Items	Example Items (During the Last Three Months, How often Have you Experienced…?)
Chronic stress screening scale	SCR	12	(Most salient items of all other scales)
External demands			
Work overload	OVRL	8	“Times when I have to postpone much needed recovery”
Social overload	SOVRL	6	“Times when I have to take care of other people’s problems too much”
Internal demands			
Occupational discontent	DIS	8	“I am missing interesting tasks that fill my day”
Lack of social recognition	RECO	4	“I don’t get enough credit for what I’m doing.”
Social isolation	ISO	6	“Times when I felt isolated from other people.”
Chronic worrying	WRY	4	“Times when my worries are over my head”

**Table 3 ijerph-17-05032-t003:** Summary of nested regression analysis for variables predicting chronic stress (SCR) and of regression analysis for variables predicting chronic worrying (WRY) (*N* = 1390).

	SCR 1	SCR2	SCR 3	SCR 4	SCR 5	WRY 1	WRY 2	WRY 3	WRY 4
Variable	β	β	β	β	β	β	β	β	β
(Intercept)	1.29 **	1.46 **	1.44 **	1.45 **	1.45 **	1.37 **	1.63 **	1.61 **	1.62 **
Sex	0.12 **	0.16 **	0.11 *	0.02	0.02	0.31 **	0.37 **	0.31 **	0.18 **
Age	0.09 *	0.08	−0.01	−0.03	−0.03	0.16 *	0.13 *	0.03	0.00
EmSt		−0.29 **	−0.22 **	−0.16 **	−0.16 **		−0.46 **	−0.37 **	−0.29 **
SocS			−0.58 **	−0.44 **	−0.44 **			−0.67 **	−0.48 **
FamS			−0.32 **	−0.20 **	−0.20 **			−0.37 **	−0.20 **
SelfE				−0.52 **	−0.44 **				−0.74 **
GSE				−0.08 *	−0.09 *				−0.09 *
EmSt × SelfE					−0.10 *				
*R* ^2^	0.009	0.046	0.307	0.425	0.428	0.029	0.084	0.296	0.436
*F* for change in *R*^2^	6.59 **	52.62 **	261.29 **	141.70 **	6.11 *	21.06 **	82.20 **	208.91 **	171.21 **

SCR = Chronic stress screening scale, WRY = stress due to chronic worrying, EmSt = employment status (reference category: unemployed), SocS = social support, FamS = family support, SelfE = self-esteem, GSE = general self-efficacy, sex reference category: male, * *p* < 0.05, ** *p* < 0.001.

**Table 4 ijerph-17-05032-t004:** Summary of nested regression analysis for variables predicting stress resulting from external demands: stress due to work overload (OVRL) and stress due to social overload (SOVRL) (*N* = 1390).

	OVRL 1	OVRL 2	OVRL 3	OVRL 4	OVRL 5	SOVRL 1	SOVRL 2	SOVRL 3	SOVRL 4
Variable	β	β	β	β	β	β	β	β	β
(Intercept)	1.39 **	1.36 **	1.35 **	1.35 **	1.36 **	1.37 **	1.46 **	1.44 **	1.45 **
Sex	0.14 **	0.13 *	0.09 *	0.03	0.03	0.02	0.04	0.00	−0.03
Age	0.11 *	0.11 *	0.04	0.03	0.03	0.12 *	0.11 *	0.05	0.04
EmSt		0.05	0.12 *	0.15 **	0.15 **		−0.15 **	−0.09 *	−0.07
SocS			−0.46 **	−0.37 **	−0.37 **			−0.38 **	−0.33 **
FamS			−0.30 **	−0.22 **	−0.22 **			−0.34 **	−0.29 **
SelfE				−0.33 **	−0.20 *				−0.29 **
GSE				−0.07 *	−0.09 *				0.10 *
EmSt × SelfE					−0.16 *				
*R* ^2^	0.012	0.013	0.201	0.252	0.258	0.006	0.015	0.18	0.209
*F* for change in *R*^2^	8.33 **	1.69	162.80 **	47.15 **	11.30 *	4.30 *	12.87 **	138.61 **	25.55 **

OVRL = stress due to work overload, SOVRL = stress due to social overload, EmSt = employment status (reference category: unemployed), SocS = social support, FamS = family support, SelfE = self-esteem, GSE = general self-efficacy, sex reference category: male, * *p* < 0.05, ** *p* < 0.001.

**Table 5 ijerph-17-05032-t005:** Summary of nested regression analysis for variables predicting stress resulting from internal demands: stress due to occupational discontent (DIS) and stress due to a lack of social recognition (RECO) (*N* = 1390).

	DIS 1	DIS 2	DIS 3	DIS 4	DIS 5	RECO 1	RECO 2	RECO 3	RECO 4	RECO 5	RECO 6
Variable	β	β	β	β	β	β	β	β	β	β	β
(Intercept)	1.55 **	1.78 **	1.76 **	1.77 **	1.77 **	1.38 **	1.50 **	1.48 **	1.48 **	1.49 **	1.49 **
Sex	−0.18 **	−0.13 *	−0.17 **	−0.23 **	−0.23 **	−0.21 **	−0.18 **	−0.23 **	−0.28 **	−0.28 **	−0.29 **
Age	0.08 *	0.05	−0.01	−0.03	−0.03	0.05	0.04	−0.04	−0.05	−0.05	−0.06
EmSt		−0.40 **	−0.34 **	−0.30 **	−0.30 **		−0.21 **	−0.14 *	−0.11 *	−0.11 *	−0.11 *
SocS			−0.47 **	−0.38 **	−0.38 **			−0.54 **	−0.46 **	−0.46 **	−0.57 **
FamS			−0.25 **	−0.16 **	−0.16 **			−0.31 **	−0.24 **	−0.24 **	−0.24 **
SelfE				−0.36 **	−0.24 *				−0.31 **	−0.19 *	−0.16 *
GSE				−0.07	−0.08 *				−0.06	−0.07	−0.06
EmSt × SelfE					−0.14 *					−0.14 *	−0.18 *
EmSt × SocS											0.094 *
*R* ^2^	0.022	0.092	0.265	0.324	0.329	0.018	0.033	0.217	0.250	0.254	0.256
*F* for change in *R*^2^	15.34 **	106.63 **	163.11 **	60.27 **	10.21 *	12.85 **	21.00 **	162.88 **	30.16 **	6.82 *	3.89 *

DIS = stress due to occupational discontent, RECO = stress due to a lack of social recognition, EmSt = employment status (reference category: unemployed), SocS = social support, FamS = family support, SelfE = self-esteem, GSE = general self-efficacy, sex reference category: male, * *p* < 0.05, ** *p* < 0.001.

**Table 6 ijerph-17-05032-t006:** Summary of nested regression analysis for variables predicting stress resulting from internal demands: stress due to social isolation (ISO) (*N* = 1390).

	ISO 1	ISO 2	ISO 3	ISO 4	ISO 5
Variable	β	β	β	β	β
(Intercept)	1.18 **	1.31 **	1.28 **	1.29 **	1.29 **
Sex	0.10 *	0.12 *	0.06	−0.02	−0.02
Age	0.12 *	0.10 *	−0.02	−0.04	−0.04
EmSt		−0.22 **	−0.13 **	−0.08 *	−0.08 *
SocS			−0.93 **	−0.82 **	−0.98 **
FamS			−0.24 **	−0.14 **	−0.14 **
SelfE				−0.42 **	−0.42 **
GSE				−0.09 *	−0.08 *
EmSt × SocS					0.14 **
*R* ^2^	0.007	0.023	0.402	0.463	0.468
*F* for change in *R*^2^	4.60 *	22.63 **	438.23 **	79.29 **	12.63 **

ISO = stress due to social isolation, EmSt = employment status (reference category: unemployed), SocS = social support, FamS = family support, SelfE = self-esteem, GSE = general self-efficacy, sex reference category: male, * *p* < 0.05, ** *p* < 0.001.

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
