# Peer review of "Associations of Social and Psychological Resources with Different Facets of Chronic Stress: A Study with Employed and Unemployed Adolescents"

_ijerph, 2020, doi:10.3390/ijerph17145032_

Round 1

Reviewer 1 Report

This in an important research that examines the impact of unemployment in adolescents stress levels and the buffering effects of social and psychological resources. Specifically, authors analyse associations between social and family support, self-esteem and self-efficacy with different stress dimensions in employed and unemployed adolescents. Data were from 1405 adolescents. Results found and stronger association between self-esteem and stress levels in employed adolescents and higher benefits from social support in unemployed adolescents. Despite I added several comments, I think that the manuscript is well written, scientifically sound and suitable for publication. However, I have identified some issues here for the authors to consider - some are minor, others more substantive. I encourage authors to revise and re-submit the manuscript. I hope they find my comments helpful.

Comments:

Abstract

  • There is some basic descriptive information about sample that is not clear either in the abstract nor the method. Please, include the age range of participants (which is not indicated in the manuscript) in the abstract and method, and where the study was conducted in the abstract (country).

Introduction

  • Authors state that the majority of adolescents in Switzerland engages in an apprenticeship after lower secondary education at age of 15 years old. From a country with a really different labour market, in which adolescent’s unemployment is much more frequent and the rates of unemployment are really higher even for adult population, the experience of unemployment it is completely different. I think that the introduction should be expanded including more detailed data that facilitates the interpretation of the results and the limitations for the generalizability of the findings to other countries, as for example, how much is the percentage of these adolescents?, which is the percentage of university students? which are the overall rates of unemployment in the country? What about the living situation of these adolescents (I assume they are still in their family houses)? How much they need those financial resources from work? Are they receiving any help for being unemployed?
  • In the introduction, authors describe previous findings about social and psychological resources in the context of stress. However, specifically when stress is related with the lack of employment and financial resources, the benefits from social support are less consistent, even for adults. I mean, there are several studies that have reported how unemployment and the lack of financial resources erodes the psychological assets and the social support. Moreover, there are also studies about how the erosion of the social capital could be a path or a mechanism through which inequalities affects health. But the most important, is that there are previous studies about the effect of social support and psychological resources in employed and unemployed both in adults and adolescents. Thus, I suggest that the introduction section should be expanded including a more specific literature about differences in how employed and unemployed people benefit differently from psychological and social support. The theory of Kawachi and Berkman, that distinguish the main effect model (social support benefit all people regardless their situation) vs the stress buffering model (more benefits of social support for people experiencing stress) could be a good starting point. This theory also serve as support for the results authors found.
  • Maybe the sentence in lines 77-78 can be added to the paragraph above.

Material, methods, and results

  • How was the sample recruited? Which were the inclusion criteria? Only apprenticeship participated in the original sample? As I understood, a second subsample was collected to complete the unemployed sample from an unemployment program, there was any difference between both groups controlling for the employment situation or between the unemployed youth from the first and the second sample? I understand that unemployed from the second subsample were actively seeking for a job, but it is unclear for me it this was the same situation for the unemployed pertaining to the initial sample.
  • Related to this issue, and regarding the unemployment situation, one of my biggest concerns with the interpretation of the results is the lack of controlling variables such as the socioeconomic status of adolescents, living situation, if they were actively looking for job, receiving any economic aid, their parental employment status… How was unemployment measured? There was no additional information about these questions?
  • What authors did exactly with the respondents that did not answer the question about their gender? How gender was measured? Was gender or sex the question? I think that authors uses gender along the manuscript, but I am not sure if the variable they measured was sex (are you a boy, girl ­­-and/or intersex-) or gender (do you identify yourself as a boy or girl, or even more, do you agree with the sex you were assigned at birth?). If the question only allowed for responding boy or girls, intersex or gender non-binary adolescents could have not responded because they don’t feel comfortable with the question itself. They were simply dropped from the analysis? If authors did this, I think that at least they should include an explanation more detailed and make a reflection in the limitations sections about it.
  • Also I think that the treatment of the sex/gender differences should be revised across the complete manuscript. In the introduction there is no reference to sex/gender differences, and in the method they stated “age and gender were entered as additional predictors of no interest into the models”. However, in the results and discussion they describe some differences among girls and boys’ experiences of stress. Given that the experience of stress, unemployment, and benefits from social support have revealed important sex differences, why authors did not included interactions by sex/gender? I understand that there are already too many results, but I think that the manuscript would benefit from simplifying this section by dropping some stress dimensions (the manuscript in its current form contains too many results and little room for interpretation), and examining more in depth these interactions by sex.
  • Authors indicated in the results that age correlated but no information is added about the meaning of the association, please, complete.
  • Authors should include a figure legend in Figure 1 indicating which is the colour for employed and unemployed lines.

Discussion

  • Again regarding the use of sex and gender, my suggestion is that authors should revise the use of gender also in the discussion section, where they refer that “female or male adolescents”…. I recommend to better clarifying what question they used, and probably, if they were measuring sex, report sex differences instead of gender differences. I suggest using gender only when the socialization process or cultural differences related to sex are reported to understand those differences. In this point, even if authors do not complete the analysis with the interactions by sex, when they report differences among boys and girls they should go further explaining possible reasons for these differences. There are also a strong body of research that have analysed stronger consequences of being unemployed for boys/men. In addition, then authors mention in lines 311-313 that girls showed more occupational stress in the work environment. I think that there is a need of a further explanation about these gender differences: how pressure, conciliation, family life or even gender gap where woman are required to demonstrated their efficiency more than men are needed.
  • In addition, the fact that unemployed youth benefit more from social support should be better explained.

  • Revise typos in line 321 and 338.

Reviewer 2 Report

Associations of social and psychological resources with different facets of stress: A study with employed and unemployed adolescents

Thank you very much for the opportunity to review this article. The authors manuscript addresses an important issue that is both theoretically and methodologically challenging. This work has the potential to contribute to understanding of the relationship between unemployment, perceived stress and other psychological variables. The exploratory study would be strengthened by clarifying the theoretical basis of the work. Methodologically, the author should provide more information on the questionnaires and on methodological limitations of the data. According to this reviewer, a comprising revision of the manuscript is necessary before publishing this work. The following remarks should support the authors revising their manuscript.

Theoretical basis

The authors should point out more clearly in how far their study adds new aspects to current research literature. They state that employment status, social and family support, self-esteem, and self-efficacy consistently predicted chronic stress. Yet, those are well known constructs and associations – the relevance of the study should be commented on more clearly. In how far does the sample differ from samples in other stress studies? At the end of the text (discussion) the authors provide more information on the relevance of their study – this should be part of the introduction.

Stress is an empirically and theoretically well established construct; the introduction of the article omits any reference to stress theories (e. g. Lazarus & Folkman, 1984, Hobfolls Conversation of Resources Theory). The link between stress and unemployment could be theoretically explained by referring to stress theories. The links between the variables should be theoretically explained- the same refers to the link between social support and stress. Anyway, the authors should state in which life domains they particularly expect unemployed adolescents to have more stress – this should be much more specific in order to contribute to the current stress literature.

Methods

The introduction and description of the instruments used is very short and insufficient. As said above, the authors do not have a consistent stress theory which their study is based on (e. g. stress a primarily biological phenomenon, stress a transactional process etc.). This limitation also has methodological consequences. The text should be more consistent here and the instruments chosen should be appropriate for the stress theory which the authors should introduce in the theory section.

Given the lack of a probability sample in this study, the author should include a more detailed description of the context, intended population and sample and the sample selection procedure.

Methodologically, the authors are doing well to test the data for systematic missings and to show that the requirements for linear regressions were met by the data. A hierarchical regression could also have been appropriate in order to differentiate between control variables like age and gender (1. Step) and the other psychological variables to be examined in the study.

Discussion

The discussion provides a comprehensive overview of the results.

“Interestingly, the perception of social support and self-esteem were more consistent and stronger predictors of stress dimensions than employment status, family support, and self-efficacy.”

This is a very interesting and new result which should be commented on in detail. In other studies socio-ecomomic factors like the employment status are highly relevant for stress (Lüdeke, 2018) – what does the authors` findings imply for the relationship between socioeconomic factors and psychological variables? This should be discussed more broadly.

Round 2

Reviewer 1 Report

Authors have addressed all the main comments and I find the manuscript suitable for publication.

Author Response

The reviewer stated that we had addressed all of his comments.

We would like, once more, to thank the reviewer for his very helpful and constructive comments in review round 1.

Reviewer 2 Report

Review: Associations of social and psychological resources with different facets of chronic stress: A study with employed and unemployed adolescents

Thank you very much for the carefully revised manuscript version. According to this reviewer the theoretical basis of the work is clarified by referring to different operationalisations, resource based stress theories and different theoretical models relating to the link between stress and unemployment. As well, the authors provide metanalytic evidence on the impact of unemployment on adolescents. The theoretical basis is much more substantial. Methodologically, the authors provide more information both on the instruments, sample selection procedure and inclusion criteria. As well, they are doing well reflecting on the limitations of their empirical study. In addition, the authors improved their data analysis by introducing a hierarchical linear regression model. Overall, the authors succeed in making their study replicable for other researches. The discussion well refers to the theoretical basis of the study. Lüdeke and Linderkamp (2018) is the study which this reviewer referred to in the first report.

The authors should partially pay more attention to formal requirements. In the abstract they now provide information on the mean age of adolescents. SD and age range should be reported at well. After careful formal revisions following the APA style this reviewer would recommend the manuscript for publication.

Author Response

"The authors should partially pay more attention to formal requirements. In the abstract they now provide information on the mean age of adolescents. SD and age range should be reported at well. After careful formal revisions following the APA style this reviewer would recommend the manuscript for publication."

We would like to thank the reviewer for his attentive lecture of our revised manuscript. We added the respective information to the abstract and revised the formal aspects of the manuscript according to APA style and the journal's formatting guidelines.

We would like to take the opportunity to thank the reviewer again for his time invested in this very constructive and helpful review process of our manuscript.